# Cancer Targeting and Diagnosis: Recent Trends with Carbon Nanotubes

**DOI:** 10.3390/nano12132283

**Published:** 2022-07-02

**Authors:** Ragini Singh, Santosh Kumar

**Affiliations:** 1College of Agronomy, Liaocheng University, Liaocheng 252059, China; singh@lcu.edu.cn; 2Shandong Key Laboratory of Optical Communication Science and Technology, School of Physics Science and Information Technology, Liaocheng University, Liaocheng 252059, China

**Keywords:** carbon nanotubes, electrochemical sensing, photoacoustic imaging, photothermal therapy, photodymanic therapy, gene therapy, immunotherapy

## Abstract

Cancer belongs to a category of disorders characterized by uncontrolled cell development with the potential to invade other bodily organs, resulting in an estimated 10 million deaths globally in 2020. With advancements in nanotechnology-based systems, biomedical applications of nanomaterials are attracting increasing interest as prospective vehicles for targeted cancer therapy and enhancing treatment results. In this context, carbon nanotubes (CNTs) have recently garnered a great deal of interest in the field of cancer diagnosis and treatment due to various factors such as biocompatibility, thermodynamic properties, and varied functionalization. In the present review, we will discuss recent advancements regarding CNT contributions to cancer diagnosis and therapy. Various sensing strategies like electrochemical, colorimetric, plasmonic, and immunosensing are discussed in detail. In the next section, therapy techniques like photothermal therapy, photodynamic therapy, drug targeting, gene therapy, and immunotherapy are also explained in-depth. The toxicological aspect of CNTs for biomedical application will also be discussed in order to ensure the safe real-life and clinical use of CNTs.

## 1. Introduction

Cancer belongs to a category of disorders characterized by uncontrolled cell development with the potential to invade other bodily organs, resulting in an estimated 10 million deaths globally in 2020 [1]. Although several areas of research have aimed to develop novel cancer treatment options, the optimum technique for decreasing cancer-related morbidity and death has not yet been reached. Nanotechnology-based systems have emerged as an alternative technique for enhancing cancer therapeutic effectiveness by using the benefits and specific capabilities of nanoparticles for drug administration, diagnostics, and imaging [2]. Several nanocarriers, including carbon nanotube (CNT), solid lipid nanocapsules, polymeric micelle dendrimers, magnetic nanoparticles, gold nanoparticles, and liposomes, are being utilized for drug delivery and enhanced cancer therapy [3]. CNTs are synthetic sp2 hybridized carbon atoms arranged in a one-dimensional hexagonal mesh with a nanosized diameter [4]. They have received a great amount of attention in biomedical applications due to its large surface area, stability, high-aspect ratio, and rich surface chemical functionalities. They have also proven to be excellent transporters for the delivery of drugs and biomolecules [5] due to their tubular and fiber-like structure and easy functionalization with nucleic acid, peptides, and proteins [6]. Through suitable functionalization, CNTs have been employed as nanocarriers to carry anticancer medicines [7], genes, and proteins for chemotherapy [8]. They have also been utilized as mediators for photothermal treatment (PTT) and photodynamic therapy (PDT) for the direct elimination of cancer cells [8,9,10]. Figure 1 shows the several advantages and disadvantages of CNTs in biomedical applications.

CNTs exhibit chirality and diameter-based physicochemical properties and are also shown to have enhanced cell penetration properties and stability. Studies have demonstrated the efficacy of CNTs in anticancer drug delivery. In this context, Joshi et al. compared the delivery of the anticancer drug methotrexate (MTX) using aminated multi-walled carbon nanotubes (MWCNTs) with aminated fullerenes as carriers in MDA-MB-231 breast cancer cells [11]. The results clearly showed that, in comparison to MWCNTs-MTX, MTX-fullerenes displayed better cytotoxic effects and higher circulation times. However, bio-corona formations of MWCNTs were significantly lower than that of the fullerenes, which resulted in enhanced bioavailability of MWCNT-MTX in comparison to MTX-fullerenes. Thus, during in vivo study, MWCNT-MTX displays better anti-tumoral effects at lower doses when compared to MTX-fullerenes. Ex vivo hemolysis results also showed the enhanced biocompatibility effect of the MWCNT-MTX composite.

Issues such as cellular toxicity, incompatibility with biological mediums, agglomeration, accumulation, and long-term persistence are some of the major drawbacks which limit the application of CNTs in real scenarios. Therefore, in the field of biomedical application, there is an urgent requirement to develop CNTs with enhanced biocompatibility. Surface functionalization, purity, surface area, and length are a few of the physicochemical properties that play a major role in governing the biocompatibility of CNTs [12]. Reports clearly showed that, in comparison to pristine CNTs, surface-functionalized CNTs are less toxic. Additionally, surface modification, via the use of cell-specific biomolecules, promotes the targeting ability of CNTs [13], thus reducing their systemic toxic effect and immunogenicity [14]. Functionalization can be achieved by linking specific functional groups to the ends or sides of CNTs via covalent or noncovalent linkage, as shown in Figure 2 [15]. Studies have shown single-walled carbon nanotubes (SWCNTs) conjugated to HPA and lectin obtained via the carboxylic-acid modifification, can be utilized for targeting MCF-7 breast cancer cells. Angiogenesis and metastasis can be induced via the alteration of protein glycosylation in a tumor microenvironment which can be detected by HPA lectin, and thus used to target cancer cells [16,17]. Due to van der Waals interactions between bundles and high surface energies, CNTs tend to form aggregates, and this inhibits their dispersion in the vast majority of organic and inorganic solvents. Therefore, the efficient implementation of CNTs requires substantial chemical changes to enhance their compatibility with host materials. Similar to functionalization, biocorona formation also significantly modifies the original CNTs complex surface characteristics (in terms of structure and function), thereby influencing their behavior in biological environments concerning toxicological properties, hydrodynamic size, aggregation, and targeting [18].

The toxicity of CNTs is highly dependent on the cell line heterogeneity, yet this can reduce cell viability in both normal and malignant cells via various mechanisms [19]. Earlier studies have shown that CNTs can promote cell death and inhibit cell proliferation. CNTs can induce oxidative stress due to generating reactive oxygen species (ROS) [20], which decreases cell adherence ability, promotes autophagic cell death [20,21], induces membrane destabilization and DNA damage [22], induces pyroptosis [23] and enhances endoplasmic reticulum stress [24]. The functionalization of CNTs greatly influences their cytotoxic effect; in this context, Zhou et al. demonstrated the in vitro cytotoxic effect of pristine and functionalized CNTs (MWCNTs-COOH and MWCNTs-OH) on A549 cells. Results showed that pristine MWCNTs cause a reduction in cell viability in comparison to functionalized MWCNTs; however, functionalized MWCNTs proved to be more genotoxic than the pristine form [25]. In vitro and in vivo results showed that NH_2_- and COOH-modified MWCNTs can significantly reduce the toxic effect in HEK293 cells and zebra fish in comparison to pristine MWCNTs [26].

To date, several outstanding articles have been published which report the use of CNTs in cancer therapy. However, none have provided a mechanism-based study of CNTs in the diagnosis and therapy of cancer. Thus, the present review article provides a deep insight into explaining the classified application of CNTs according to their mode of sensing and therapy in cancer cells. Moreover, we have also discussed the toxicological aspects of CNTs regarding their application for biomedical purposes. Figure 3 shows the evolution of CNTs in the diagnosis and treatment of cancer.

## 2. Application of Carbon Nanotubes in Cancer Diagnosis

This section describes the various sensing techniques for cancer detection. These are primarily based on the utilization of various CNTs.

### 2.1. Electrochemical Sensing

In addition to cancer, CNTs are widely employed in the diagnosis of several other diseases, such as Parkinson’s disease, which is a common neurodegenerative disorder caused by a lack of dopamine in the brain; the presence of levodopa can be used to detect it. Its increased concentration causes a patient’s movement to become disordered, as well as facilitating uncontrollable emotions. This, as well as the detection of levodopa, is extremely important. As shown in Figure 4, Ji et al. developed a smartphone-based electrochemical sensor that can quickly detect levodopa [27]. This sensor is made up of a smartphone, a hand-held electrochemical detector, and a disposable carbon nanomaterials sensor. The sensor uses functionalized screen-printed electrodes with gold nanoparticles (AuNPs) and single-walled carbon nanotubes (SWCNTs) to monitor electrochemical current signals in the presence of levodopa molecules. The electrochemical detector’s job here is to convert electrochemical excitation signals into currents. Following that, a smartphone is connected to the detector, which aids in controlling the detector and obtaining data, plotting the graph in real-time. This differential pulse amperometry-based sensor can detect 0.5 milligrams of levodopa in a human serum sample. It is an electrochemical sensor that has high selectivity and can quickly distinguish levodopa from other representative substances in the body. Overall, it is a sensitive and selective sensor that can be used in point-of-care testing.

To improve sensing performance, the electrode is functionalized with AuNPs and SWCNTs. The electrochemical reaction of the proposed levodopa sensor is shown in Figure 5a. Levodopa reacts with the SWCNTs and AuNPs to activate the surface of the electrode, which is then oxidized and produces the electron for the electrochemical reaction. SEM images of the AuNP/SWCNT/chitosan-coated electrode are shown in Figure 5b,c. According to these images, the graphite coating on the electrodes appears to be rough, yet the surface of the AuNP/SWCNT/chitosan immobilized electrode appears to be a net structure, with AuNPs clinging to the SWCNTs. The stability of the biosensor is improved by this functionalization. The SWCNTs also increase the active surface area of electrodes in this case. The electrochemical performance in the presence of various modified electrodes is compared in Figure 5d. It can be seen that the bare electrode has the lowest response and that, as functional layers are added, the response increases. This result also demonstrates that AuNP-SWCNTs can improve a screen-printed electrode’s sensing performance. However, a composite of AuNPs and SWCNTs improves the electrode’s electrochemical performance, conductivity, and electrocatalytic properties. Due to unique advantages such as fast electron transfer kinetics, a wide electrochemical stability window, and large length-to-diameter aspect ratios, SWCNTs are widely used in photovoltaic applications, electrolytic water, and biosensing applications. As a result, SWCNTs can be used as a potential nanoscale building block for subsequent nanosensor fabrication [27].

Thereafter, Fan et al. developed a smartphone-based electrochemical sensor for detecting the cancer antigen 125 (CA125) using a differential pulse voltammetry (DPV) measurement scheme [28]. As is well-known, CA125 is a crucial tumor marker which is frequently found in ovarian cancer, breast cancer, lung cancer, and other related diseases. Due to the size of the instruments, former hospital testing for CA125 was limited. This prevented the portability and availability of testing in remote areas. Concerning point-of-care applications, this group have optimized a sensor with features including portability, low cost, high accuracy, and small size. As is well known, smartphone demand has exploded over the past several years. This category of point-of-care testing devices has been used to identify various analytes, including sulfane sulfer, levodopa, human immunodeficiency virus (HIV), secretory leukocyte protease inhibitor (SLPU), biomarkers, blood-ketone, uric acid, and white blood cells [27,29]. This detection method replaces cumbersome and costly hospital instruments, simplifies the measurement steps, and shortens the detection time. As depicted in Figure 6, the authors of this study have developed a CNT-assisted smartphone-based electrochemical sensor for CA125 detection.

Bluetooth is used to connect this immunosensor-based electrochemical sensor to a smartphone. The role of the smartphone is to provide a user-friendly interface and facilitate internet-based communication with a remote medical center. The obtained results were also compared using Roche electrochemical luminescence immunoassay (ECLIA) tests for clinical validation. In this instance, an Android smartphone, a screen-printed immunosensor, and an electrochemical detector were used to develop a smartphone-based electrochemical sensor. Nanocomposites of multi-walled carbon nanotube/thionine/gold nanoparticles (MWCNT/Thi/AuNPs) are used to fabricate the screen-printed immunosensor, with an induced redox reaction in Thi, in this instance. After adding the sample to the immunosensor surface for detection, the CA125 antigen reacts with the CA125 antibody (anti-CA125) to form an immunocomplex. This nonconductive immunocomplex inhibits the transfer of electrons from Thi. The calibration curve demonstrates that DPV current decreases with increasing concentrations of CA125. Furthermore, Lv et al. showed the immobilization of bimetallic rhodium@palladium core-shell nanodendrites (Rh@Pd ND) over sulfate-activated MWCNTs and used this as an electrochemical immunosensor for CEA detection [30]. Rh@Pd ND exhibits multiple catalytic sites and an enhanced surface area, high solubility, and electrical conductivity. In optimal conditions, this sensor can detect CEA within the linear range of 25 fg/mL–100 ng/mL. Cyclodextrin modification can also be used to improve the immobilization performance of CNTs. Additionally, carbohydrate antigen (CA) exhibits a strong correlation with cancer, as its elevated level increases the risk of tumor progression [31]. CA199 can be detected by electrochemical immunosensor composed of antibody-conjugated MWCNT-Fe_3_O_4_ dispersed in chitosan. The detection limit has been calculated to be 0.163 pg/mL, with a linear detection range from 1.0 pg/mL–100 ng/mL [32].

CNT-based thin-film transistors (TFTs) have also demonstrated great promise in biosensing applications, particularly for the development of label-free and highly sensitive DNA detection. However, defects in the dielectric and channels of CNT-based TFTs inevitably occur and can result in noisy or unreliable output signals. Thus, Ren et al. [33] proposed a method for enhancing the performance of TFT-based biosensors by employing a low-temperature supercritical carbon dioxide (SCCO_2_) fluid activation method. They developed a cell-free DNA sensing platform using an all-CNT-functionalized peptide probe. This type of sensor offers a promising and universal strategy for achieving highly accurate and sensitive transistor-based biosensors for future clinical applications.

### 2.2. Immunosensing

Normally, immunosensing is based on the typical processes for signal-off assays. Farzin et al. [34] proposed a simple and label-free voltammetric immunosensor for the rapid detection of prostate specific antigens (PSAs). PSAs are a kind of single-chain glycoprotein secreted by the epithelial cells of the prostate gland, normally found in a human serum. The concentration of PSAs increases during cancer; thus, it is used as a noninvasive biomarker. The typical PSA concentration in human serum during prostate cancer is 4 ng/mL. Till now, several immunosensing technologies, such as fluorescence immunosensors [35], electrochemical immunosensors [36], surface-enhanced Raman scattering-based immunoassays [37], and surface plasmon resonance (SPR) immunosensors [38] are used for PSA detection. Farzin et al. [34] proposed an immunosensor based on the multi-walled carbon nanotube (MWCNT)/L-histidine immobilized reduced graphene oxide (His-rGO) for attaching a thionine redox indicator and anti-PSA antibody (Ab). MWCNTs play a crucial role in the facilitation of electron transfer between thionine and the glassy carbon electrode and the enhancement of electrical conductivity.

In another study, Ding et al. demonstrated the antibody-activated, vertically aligned carbon nanotubes array (VANTA) for the detection of oncoprotein CIP2A, which is involved in various cancers such as breast, oral, and multiple myeloma cancers [39]. VANTA coating has attracted much attention due to its unique properties of electrical conductivity, light absorption, and chemical inertness. A developed sensor showed a detection limit of 0.24 pg/mL, with a linear range from 1–100 pg/mL in saliva. This sensor exhibits the highest sensitivity in comparison with the CIP2A enzyme linked immunosorbent assay. Thus, this sensor paved the way for rapid and early screening for the detection of oral cancer. In another study, Soares et al. also developed an MWCNT-based immunosensor for the detection of the pancreatic biomarker CA19-9 [40]. The sensor also consists of nanostructured mats of electrospun nanofibers of polyamide 6 and poly(allylamine hydrochloride). Results showed a detection limit of 1.84 U/mL. The high sensitivity of the sensor may be attributed to irreversible adsorption between the antigen and the antibody. This was further confirmed by polarization-modulated infrared reflection absorption spectroscopy. The sensor has also tested real samples of patients’ blood serum with distinct concentrations of CA19-9. Results showed accurate detections with interference from analytes present in the biological fluids. Thus, it can be treated as a powerful, effective, simple, and accurate technique for detecting early-stage pancreatic cancer.

### 2.3. Photoacoustic Imaging

Photoacoustic (PA) imaging is a highly effective method comparable to ultrasonography. In this instance, the output signal for PA imaging is also a broadband acoustic wave, but the source is a light that induces a region of tissue to become an active acoustic source [41]. Some tissues exhibit intrinsic PA properties, but the majority of diseases do not, necessitating the use of an exogenous contrast agent. In a similar manner, Pristine CNTs exhibit PA properties and are widely used to enhance PA properties during sensing. Typically, it is necessary to employ double CNT functionalization depicted in Figure 7A. The first function was indoCyanine green (ICG) dye through-stacking interactions to enhance PA performance by increasing optical absorption, as depicted in Figure 7B. The second method involved attaching cyclic Arg-Gly-Asp (arginylglycylaspartic acid–RGD) peptides to the surface of PEGylated CNTs in order to target αVβ_3_ integrins, which are overexpressed in tumor vasculature. Along with the control (nontargeted peptide RAD that did not bind to αVβ_3_ integrins), these were analyzed. Reports showed that a coating of gold over SWCNTs enhances their inherent PA signal; therefore, some researchers have developed “golden nanotubes” (GNTs) [42].

MWCNTs and SWCNTs with strong NIR absorbance act as photothermal agents. This strong NIR absorbency makes nanotubes an excellent medium of contrast for PA imaging. Various reports showed the in vitro and in vivo application of SWCNTs in PA imaging. In comparison to the control, SWCNTs can offer more than a two- and six-fold signal amplification in thermoacoustic tomography and photoacoustic tomography, respectively. SWCNTs provide the highest contrast signal in comparison to other carbon nanomaterials, graphite, and fullerenes, which makes them an ideal contrast medium candidate for PA imaging [42].

Thus, it can be concluded that CNTs exhibiting high NIR absorbance prove to be an excellent contrast agent for PA imaging. Additionally, CNTs combined with different absorptive nanomaterials constitute enhanced or multiplexed PA imaging. Thus, various CNT-based PA-imaging probes depend on SWCNTs and MWCNTs for a highly enhanced imaging technique. Figure 7C depicts the PA images functionalized with CNTs over the standard ultrasound images. According to Figure 7C, there is a linear correlation between the CNT concentration and the respective PA signal.

### 2.4. Fluorescence Imaging

Fluorescence imaging plays a crucial role in medical diagnosis; however, a low penetration depth limits their wider application. In order to overcome this problem, researchers have developed advanced fluorescence probes which can be excited at wavelengths near to the biological transparent NIR window [42].

In this context, CNTs offer a large surface area and versatile surface chemistry for the immobilization of multiple active centers. This forms new pharmacological complexes for these reasons. González-Domnguez et al. chemically functionalized the SWCNTs with drugs such as fluorescein, folic acid, and capecitabine in order to develop a fluorescence sensor for colorectal cancer [44]. NCC successfully stabilizes functionalized SWCNTs in water dispersion, with the resulting hybrids exhibiting no toxicity, in contrast to surfactants such as DOC, and a stabilization ability comparable to that of other polymers and biomolecules, such as PEG, GG, and ALB. Aqueous dispersions of fluorescein-functionalized SWCNTs in type II-NCC demonstrate an increase in the hybrid’s intrinsic activity against colon cancer cells compared to nonfunctionalized counterparts and the standard drug capecitabine while being nontoxic to normal colon cells. Generally, synthesized SWCNTs did not exhibit fluorescence activity excited under specific wavelengths and thus produce dark field images. With the internalization of polarization purified SWCNTs, most nanotubes show a significant decrease [42].

Fluorescence imaging enables the complete in-depth removal of a tumor at a microscopic level. Ceppi et al. demonstrated a reflectance/fluorescence imaging system for ovarian cancer in mice to both quantify the tumor as well as evaluate the postoperative survival, guided by fluorescence image surgery. In this study, a contrast agent was composed of SWCNTs conjugated to an M13 bacteriophage carrying a peptide specific to the SPIRC protein (protein overexpressed extracellularly in ovarian cancer) [45]. The developed imaging system can detect second near-infrared window fluorescence (1000–1700 nm) and helps in intraoperative tumor debulking by displaying real time video. The authors found increased survival in animals treated with fluorescence image-guided surgical resection in comparison to standard surgery. In another study, Lee et al. developed the platelet-derived growth factor (PDGF) aptamer with conjugated SWCNTs, based on an NIR optical sensor [46]. Results showed significant change in NIR fluorescence of the SWCNTs due to the conformational change in the aptamer, which reversibly regulates the refolded aptamer functionalized SWCNTs NIR fluorescence. In another study, Zhang et al. demonstrated a nanocomposite consisting of MWCNTs and magneto fluorescent carbon quantum dots for the dual-modal imaging of cancer cells in mice [47].

### 2.5. Raman Imaging

As was previously mentioned, the fluorescence technique is widely applicable to multiple biological imaging applications. However, nonideal factors such as the autofluorescence background from biological tissues, the photobleaching of organic dyes, and the wide fluorescence excitation and emission peaks that result in spectral overlaps limit the use of multiple colors in an experiment. Therefore, scientists have developed Raman scattering, which has narrow spectral lines and is useful for imaging within a high multiplicity. Raman spectroscopy has been widely used for biomedical diagnostic purposes, as it provides key information regarding the chemical composition of cells and tissues. SWNTs exhibit different Raman peaks, including tangential G band (~1580 cm^−1^) and radical breathing mode (RBM 100–300 cm^−1^) [42]. Distinguishing SWCNTs Raman peak from the autofluorescence background can be easily done due to narrow and sharp peaks. SWCNTs Raman excitation and scattering photons can reach the NIR region for in vivo imaging.

In addition, tissue autofluorescence issues can be avoided because sharp Raman peaks can be distinguished from the fluorescence background. Raman excitation can also be selected in optical windows with low background and biological transparency. Raman imaging looks promising for the future of biological imaging [48]. SWCNTs and other novel nanomaterials are widely used in a variety of biological applications, such as drug delivery and imaging, as is well known. SWCNTs also possess intrinsic optical properties required for biological imaging, including strong resonant Raman scattering and photluminescence in the NIR range.

### 2.6. Colorimetric Sensing

Several optical sensing techniques, such as chemiluminescence, fluorescence, and colorimetric, are capable of detecting various types of cancer biomarkers. There are a number of unique advantages to colorimetric sensing, such as low costs and dispensing with complex tools and specialized personnel [49]. Color changes are visible to the naked eye, so sophisticated tools are not required for data analysis. The prominent physical, chemical, and optical properties of carbon-based nanomaterials, such as graphene oxide, graphene quantum dots, and CNTs, attract researchers today, and their clinical applications are intensively studied [50].

### 2.7. Plasmonic Sensing

Surface plasmon resonance (SPR)-based sensors are widely used in chemical and biosensing applications due to their unique properties, such as high sensitivity and label-free sensing. During optical simulation, surface plasmon waves are generated at the interface between a metal layer, such as gold or silver, and a dielectric layer. In this instance, the change in refractive index can reveal the molecular binding on the metal surface [51]. Lisi et al. proposed a plasmonic sensor for tau protein detection based on surface plasmon resonance coupled to carbon nanostructures [52]. They have also elaborated on the process of amplification caused by CNTs, as shown in Figure 8. Here, MWCNTs are used as mass enhancers, following their conjugation to the secondary antibody in order to amplify the SPR signal. To the best of our knowledge, there exists no CNT-based plasmonic sensor for cancer detection that is currently available. This may be the scope of future work. Table 1 summarizes the CNT-based cancer detection techniques.

## 3. Application of Carbon Nanotubes in Cancer Therapy

CNTs have been extensively studied as carriers for the delivery of various agents, such as contrast media and therapeutic agents like drug-based, nucleotide-based, and plasmid-based CNT complexes, as well as their synergistic effect in combination with cytotoxic agents. This has the potential for forming hybrid systems in conjunction with various techniques, such as phototherapy and sonodynamic therapy, for combination treatments of cancer. Almost all of this research implies that CNTs might be considered for unique techniques for early cancer detection and enhanced cancer treatment. Table A1 in Appendix A summarizes the various carbon-based nanomaterials for their application in cancer therapy.

### 3.1. Drug Targeting

Chemotherapy has been used in cancer treatment in combination with other treatments like surgery and radiation. However, limitations like nonspecific drug release and toxic side effects can increase drug resistance and confine the therapeutic window, respectively. Moreover, other limitations are the susceptibility of drugs towards enzyme degradation, and denaturation, which can alter their in vivo efficacy. Therefore, the emergence of new techniques for specific drug targeting with reduced toxicity and enhanced therapeutic efficacy will be needed [1,70]. In order to solve this issue, nanomaterial-based delivery systems came into existence and offered solutions to most of these issues. Nanomaterials can accumulate around tumor sites by enhanced permeation and retention (EPR) effects due to leaky blood vessels and the absence of lymphatic drainage in tumor tissue. Furthermore, encapsulation of drugs into nanomaterials provides them with a protective layer from the surrounding environment as well as also enabling the extended release of drugs from nanocarriers. CNTs make for an ideal candidate for drug delivery due to the following properties: high loading capacity, pH-dependent sustained drug release, enhanced cellular internalization, large surface area, high aspect ratio, and stability and modification capability [5]. By utilizing the benefits of CNTs, Liu at al. developed the doxorubicin delivery system using PEG-functionalized SWCNTs to reach a loading capacity of ~400% by weight [71]. Researchers showed that the π-stacking of aromatic molecules mainly depends on nanotube diameter and this enables a controlled release rate of drug molecules. Additionally, pH is another important factor that governs drug-loading capacities, as it can have significant effects on the ionization degree and the free site for adsorbents. Due to having large surface areas, SWCNTs have been shown to exhibit a higher drug loading capacity in comparison to MWCNTs [72], whereas, in comparison to long CNTs, the smaller of the two shows a higher loading capacity achieved in less time [73]. Moreover, aromatic-content-carrying peptides show a higher binding affinity to SWCNTs due to their interaction with the π electrons of SWCNTs [74]. In another study, Yang et al. demonstrated COOH, PEG, and PEI functionalized SWCNTs as DOX carriers for the treatment of MCF-7 cells [75]. Results showed that SWCNT-PEG-PEI exhibits the highest anti-tumor effect and drug delivery potential under acidic conditions in comparison to CNT-COOH and CNT-PEG (Figure 9). Flow cytometry and fluorescence-based studies indicate the enhanced internalization of SWCNT-PEG-PEI which promotes tumor cell death via apoptosis. These advantages may be attributed to their high dispersibility and greater affinity towards cancer cells.

The susceptibility of CNTs to release drugs in the acidic environment of tumor cells enables passive targeting of the tumor site. Thus, the application of CNTs in the pH-dependent release of anticancer drugs is another major advantage of their use in cancer drug delivery systems. It reduces side effects, increases drug circulation time and specificity, decreases administration frequency, and preserves the optimum drug concentration [76]. In this context, Gu et al. developed the DOX and hydrazinobenzoic acid (HBA) functionalized SWCNT complex via hydrazine bonding and evaluated its cytotoxic effects on HepG2 cancer cells [77]. Results showed that the complex exhibited a pH-dependent drug release rate, with maximum release occurring at a lower pH (5.5) (tumor cell pH) in comparison to a higher pH (7.4). After 60 h of incubation, only 50% of the drug was released from the SWCNT-DOX composite in comparison to the SWCNT-HBA-DOX complex (73% drug release), indicating the high stability of π–π stacking interactions in comparison to hydrazone bonding. Results also showed the higher cytotoxic effects of SWCNT-HBA-DOX in comparison to the SWCNT-DOX complex due to enhanced cellular internalization. Furthermore, Cao et al. demonstrated that the PEI modified MWCNTs covalently conjugated to hyaluronic acid for the targeted delivery of DOX to the cancer cells overexpressing CD44 receptors. Authors showed that the synthesized complex exhibited a drug loading capacity of 72% and released at a higher rate in acidic pH (5.8 pH in cancer) in comparison to physiological conditions (pH 7.4) [78]. Results showed that MWCNT-HA-DOX showed good biocompatibility in the tested concentration range and also exerted significant toxic effects to cancer cells (Figure 10).

Cancer cells require a high amount of folate for DNA synthesis and rapid proliferation; therefore, the folate receptor is found to be overexpressed in many cancer cells. Thus, conjugation with folic acid (FA) is another commonly used approach for targeting cancer cells [79]. In this context, Lu et al. developed a magnetic dual-targeted nanocomposite composed of MWCNTs and IONPs for drug delivery purposes [80]. MWCNTs were functionalized with poly(acrylic acid) via free radical polymerization and conjugated to FA for DOX drug loading. This nanocomposite utilizes dual-targeting effects via magnetic field and ligand-receptor interaction. In comparison to free DOX, DOX conjugated to the MWCNTs with high efficiency via hydrogen bonding and π–π stacking exhibited enhanced cytotoxic effects towards U87 human glioblastoma cells. Furthermore, results showed that the synthesized nanocomposite had been efficiently internalized by the cells, which induced the intracellular release of DOX, and also the nanocomposite can be transported to the nucleus along with the nanocarrier. Thus, the developed nanocomposite can serve as an efficient tool for the targeted delivery of anti-cancer drugs in cancer therapy.

### 3.2. Photothermal Therapy

Photothermal therapy (PTT) is a minimally invasive therapeutic technique that leads to the thermal ablation of cancer cells via focusing on creating local heat using an optical absorption agent, also known as a photosensitizer, that can absorb electromagnetic energy and convert it to heat [81]. Hyperthermia has been demonstrated to significantly enhance tumor eradication by boosting immune activation and generating long-term immunity against metastatic tumors [82]. Traditional photosensitizers suffer from several disadvantages, such as adverse effects on the skin, reduced tumor targeting, and a limitation of therapeutic effects in hypoxic environments. CNTs exhibit superior photophysical properties, such as a broad electromagnetic absorbance spectrum and the conversion of near-infrared I and II windows that correspond to the optical transmission window of biological tissues, as well as greater target-site accumulation, proving them to be next-generation photosensitizer agents for the effective PTT of cancerous cells [83]. In this context, Sobhani et al. demonstrated the PTT of HeLa and HepG2 cells using PEG-wrapped CNTs (to improve the dispersibility of the CNTs) [84]. Thermogravimetric analysis confirmed the presence of 80% (*w*/*w*) PEG over the CNT surface. Furthermore, authors evaluated the effect of PEG-CNTs on reducing melanoma tumor size by exposing the tumor-bearing mice to a continuous-wave near-infrared laser diode for 10 min once during the treatment. Results showed a significant reduction in tumor size of the mice receiving CNT-PEG with laser irradiation in comparison to mice receiving the laser radiation alone. Thus, it can be predicted that CNT-PEG serves as an efficient tool for the eradication of solid tumors via the PTT technique. Another study also showed the photothermal effect of a hybrid complex composed of MWCNTs and gold nanostar on B16F10 mouse melanoma cells [85]. This synthesized nanocomposite did not contain any surfactant during synthesis, thus, it was biocompatible, cellular friendly, and had no significant toxicity even at higher concentrations. Results showed that under 808 nm irradiation, the developed hybrid exhibited a photothermal effect that was 12.4% and 2.4 times higher in comparison to gold nanostar alone and gold nanospheres, respectively. Also, even at a lower concentration of 0.32 nm, the developed nano-hybrid showed enhanced photothermal efficiency in killing cancer cells. In another study, McKernan et al. developed a SWCNT-based new strategy for the treatment of metastatic breast cancer cells by combining the effects of PTT and immunostimulation using annexin A5 (ANXA5)-conjugated SWCNTs and an anti-CTLA-4 check point inhibitor [86]. ANXA5 acts as a tumor-targeting protein and reduces the amount of SWCNTs required to eradicate the primary tumor. The authors demonstrated that the combined treatment leads to a significant increase in the survival rate of mice, as combinatorial treatment increases the number of CD4^+^ helper and CD8^+^ cytotoxic T cells. An increase in T cells leads to an abscopal response, where antitumoral effector cells suppress tumor metastasis. Results showed that SWCNTs can be present in organs even 4 months after the initial administration, having no toxic effects during the experiment. Furthermore, Zhao et al. demonstrated the combined synergistic effect of PTT and gene therapy in anti-tumor activity [87]. However, the targeted and controlled release of genes still remains a major challenge. Researchers coated the SWCNTs and MWCNTs with peptide lipid and sucrose laurate to form a bifunctional delivery system with enhanced photothermal effects and temperature sensitivity. Results showed that the CNT/siRNA conjugate can silence the surviving expression and thus effectively suppress tumor growth, and exhibits photothermal effects upon NIR exposure. Coating of peptide lipid and sucrose laurate facilitate the phase transition of lipids and thus enables the systemic delivery of siRNA to the tumor site (Figure 11). SWNTs produced by the CoMoCAT^®^ method are known to be highly enriched with nanotube chirality and exhibit an absorption band at 980 nm. Its photothermal activity has also been widely used for selective phototissue interaction [86,88]. Zhou et al. conjugated the CoMoCAT^®^-SWCNTs with folate, which can effectively bind to the folate receptor present on the tumor’s surface [89]. In vitro and in vivo investigations showed that FA-SWNT significantly reduced the photothermal destruction of nontargeted normal cells while significantly enhancing the photothermal death of tumor cells.

Fibrin is a final product of coagulation response and found in abundant quantity at the vasculature injury site due to the positive feedback property of coagulation and signal amplification responses [90]. Fibrin found in the tumor vessel can act as a therapeutic target for drug delivery due to its easy accessibility, ubiquitous presence, and high expression level [91]. In this context, Zhang et al. took advantage of fibrin amplified formation and developed the CREKA conjugated MWCNT-PEG nanosystem for the PTT of cancer cells [92]. Here, PEG acts as shelter, with the CREKA peptide having a higher affinity for fibrin and thus serves as a targeting agent. In vivo results showed that the exposure of MWCNT-PEG significantly increased the temperature of the tumor region 24 h post NIR irradiation. It was shown that IR783-labeled CMWNT-PEG with illumination had accumulated in the tumor tissues by an amount 6.4 times greater than that of the control group and was considerably higher than other treatment groups. It was discovered that in vivo dispersion of Cy3-labeled CMWNT-PEG was considerably greater than in tumor slices. The CMWNT-PEG almost completely eradiated the tumor xenografts after four illuminations. Overall, CMWNT-PEG demonstrated significant tumor targeting and photothermal therapeutic effectiveness. The use of CNT-based tumor-targeting conjugates, in combination with photothermal treatment, can result in more precise and efficient tumor elimination. MWCNTs coupled to a Pgp-specific antibody (Pab) were employed by Suo et al. for photothermal ablation of P-Glycoprotein (Pgp)-mediated multidrug resistant NCI/ADR-RES ovarian cancer cells. Results showed a significantly higher internalization of the Pab–MWCNTs in 3T3-MDR1 in comparison to 3T3 cells at different time periods. Upon NIR irradiation, Pab–MWCNTs exert dose-dependent specific photokilling in 3T3-MDR1 when compared to 3T3 cells.

### 3.3. Gene Therapy

Gene therapy is a new method of cancer treatment that addresses the gaps in current treatments. Gene therapy and its synergistic combination with chemotherapeutics is gaining popularity in cancer treatment. The genes of interest are transfected into the chosen cells or employed to compensate for the cells’ deficiencies in gene therapy. Poor transfection success and insufficient endosomal escape of the genes from nanocarriers, however, limit the therapeutic applicability of nanocarriers (siRNA) [1]. In this context, Cao et al. developed novel pH-responsive SWCNTs functionalized with PEI-betaine and further modified using the peptide BR2 for the co-delivery of DOX and survivin siRNA [93] (Figure 12). PEI and betaine were covalently conjugated, and the nanocomposite was synthesized with oxidized SWCNTs to form SWCNT-PB (SPB). Results showed that BR2 can be effectively internalized into HeLa and A549 cancer cells, but no internalization has been observed in 293T normal cells. SPBB-siRNA showed less survivin expression and a higher apoptotic index than Lipofectamine 2000. siRNA/DOX can be released into the A549 cell’s cytoplasm and nuclei without lysosomal retention. In comparison to SPBB-siRNA, or SPBB-DOX treatment alone, exposure to SPBB-DOX-siRNA showed synergistic effects and exhibited a significant reduction in the tumor volume of A549 cells in nude mice. SPBB-DOX-siRNA may have therapeutic benefits on tumors without causing harm to normal tissues, according to pathological studies. Finally, the novel functionalized SWCNTs loaded with DOX and survivin siRNA were successfully synthesized, and the nanocomplex exhibited effective antitumor effects both in vitro and in vivo, indicating that the nanocomplex could be used as an alternative strategy for the delivery of antitumor drugs and genes.

Suicide gene therapy is widely regarded as among the most successful approaches in the field of gene therapy. In this method of therapy, therapeutic transgenes are used either to express a toxic product from a toxic gene or transform a nontoxic prodrug into a toxic one. Both of these processes are carried out in order to combat the effects of disease. This method was utilized in the treatment of several forms of cancer, such as breast [94], liver, colon [95], prostate [96], glioma [97], and lung cancer [98]. Recent research has shown that this strategy is successful in treating chemo-resistant cancer cell lines [99] and also enhances the effectiveness of radiation during therapy [100]. In this context, Dargah et al. demonstrated that *iC9* gene induces apoptosis in MCF-7 human breast cancer cell lines [101]. In this study, pyridine modified MWCNTs were used as carriers to transfer the gene of interest. Results showed that MCF-7 cells were significantly eliminated by this approach, and in combination with chemotherapeutic approaches, it can also pass cell cycle arrest. Thus, research showed that delivering *iC9* suicide gene therapy through pyridine functionalized MWCNTs is an effective method for killing cancer cells. Furthermore, our findings demonstrate that combination therapy works even better than monotherapy because it is able to circumvent the cell cycle arrest caused due to chemical drugs. Zhang et al. demonstrated the fluorescent carbon nanoparticle (FCN)-based siRNA conjugate (C-siRNA) for the gene regulation and the treatment of cancer [102]. C-siRNA consists of chitosan-derived FCN as a core and siRNA as a shell, which can down-regulate the polo-like kinase-1 expression, which is a key regulator of mitosis in cancer therapy. In comparison to AuNPs, only one-thirtieth of the concentration of FCN was required for transfer of same amount of siRNA. In comparison to commercial nonviral gene delivery vectors, i.e., Lipofectamine 2000, C-siPlk1 treatment induces 31.9% and 20.33% apoptosis in A375 and MCF-7 cells, respectively. After intravenous administration of C-siPlk1 to mice carrying the A375 tumor, the volume of the tumor reduced to less than one-eleventh of that in the control groups. Therefore, C-siRNAs have the potential to be extremely useful agents for gene transport as well as gene therapy.

### 3.4. Immunotherapy

The term “immunotherapy” refers to a therapy which can improve the body’s ability to fight against illness, via either the modification or amplification of the immune system [103]. Antigenic targets have been utilized in effective defensive strategies to improve therapeutic efficacy against chronic infectious illnesses and cancer, as well as the blocking of regulatory systems that might hamper immunotherapeutic effects. Immuno-based oncotherapy has been efficiently utilized in the treatment of cancer. Xia et al. demonstrated a MWCNTs-based nano-delivery system containing unmethylated CpG motifs, oligodeoxynucleotide, and H3R6 polypeptide (MHR-CpG) for prostate cancer immunotherapy [104]. In vivo and in vitro results showed that the developed MHR system exhibited enhanced biocompatibility and targeted endosomal TLR9. Additionally, the utilization of MHR improved the immunogenicity of CpG in both the humoral and the cellular immune pathways. This was demonstrated by an increase in the expression of CD4+ T-cells, CD8+ T-cells, TNF-, and IL-6. The in vivo anti-cancer investigation on RM-1 tumor-bearing mice demonstrated that MHR-CpG has the ability to deliver immunotherapeutics to the tumor site as well as lymph nodes that inhibit the tumor development. Based on these findings, it appeared that MHR-CpG was a potentially useful multifunctional nano-system for the immunotherapy of prostate cancer.

### 3.5. Photodynamic Therapy

Photodynamic treatment (PDT) is a noninvasive, low-toxicity method of phototherapy that uses light and a photosensitizer chemical in combination with molecular oxygen to induce cell death [1]. After a photosensitizer is applied topically or intravenously to cancer cells, the photosensitizer is activated by specified wavelengths of light, such as NIR, resulting in energy transfer cascades that produce ROS and induce selective cytotoxicity against malignant cells [105]. In this regard, Racheal et al. synthesized zinc phthalocyanine-spermine-SWCNTs and compared them to ‘free zinc mono carboxy phenoxy’ phthalocyanine and ‘zinc mono carboxy phenoxy’ phthalocyanine-conjugated spermine in terms of photophysical properties and PTT performance against MCF-7 breast cancer cell lines [106] (Figure 13A). When compared to free zinc phthalocyanine, they discovered that both ZnMCPPc-spermine and ZnMCPPc-spermine-SWCNT had enhanced photophysical characteristics, with over 50% improvements in triplet and singlet oxygen quantum yields. In vitro cytotoxicity testing on MCF-7 cancer cells revealed that the PDT impact of ZnMCPPc-spermine resulted in a 97% reduction in cell viability at 40 mM, whereas ZnMCPPc-spermine-SWCNT resulted in a 95% reduction. In another study, Racheal and Nyokong investigated the photodynamic treatment effects of a modified zinc phthalocyanine-SWCNTs in combination with ascorbic acid on MCF-7 cancer cells [107]. The photophysical characteristics and the PTT effect of ZnMCPPc, ZnMCPPc-AA, ZnMCPPc-SWCNT, and ZnMCPPc-AASWCNT were compared in this work. Results showed that ZnMCPPc-SWCNTs had improved photophysical characteristics, enhanced lifetimes and quantum yields, and improved singlet oxygen quantum yields than ZnMCPPc alone. In addition, ZnMCPPc-SWCNT showed the best PTT impact on MCF-7 cancer cells, resulting in a 77% reduction in cell viability. The comparison showed that the conjugation of SWCNTs can boost the PTT activity of ZnMCPPc, yet not as much as spermine.

Sundaram et al. demonstrated hyaluronic acid- and chlorin e6- conjugated SWCNTs and evaluated the effect of the nanobiocomposite’s PDT on colon cancer cells [108] (Figure 13B). The hyaluronic acid coating can efficiently enhance the dispersibility of the nanocomposite. In comparison to free Ce6, the synthesized SWCNT composite shows enhanced anticancer effects on Caco-2 cells. This SWCNT composite exhibits a high surface area and strong binding; thus, the synthesized nanobiocomposite demonstrated an enhanced ability to deliver photosensitizer and high apoptotic activity in colon cancer cells [70]. Flow cytometry results indicate that the combined effects of nanobiocomposite and PDT (both 5 and 10 J/cm^2^ laser irradiated) significantly enhance cells in both early and late-stage apoptosis in comparison to control cells. A detailed examination of this study found that 10 J/cm^2^ (41.9, 6.65) had higher cell counts in early-stage apoptosis than 5 J/cm^2^ (36, 6.4) exposure. Similarly, live cell counts revealed that 10 J/cm^2^ (53.4, 6.8) had a lower percentage than 5 J/cm^2^ (58.27, 5.9). In comparison to free Ce6 and empty SWCNTs, SWCNT-HA-Ce6 with 10 J/cm^2^ laser irradiation demonstrated higher apoptotic activity. Shi et al. demonstrated the hyaluronic acid-conjugated CNTs derivative had properties like tumor targeting and enhanced solubility. Furthermore, CNTs were transformed with a new PDT agent (hematoporphyrin monomethyl ether (HMME)) to synthesize CNT-HA-HMME. This nanocomposite had the ability to combine local selective photodynamic therapy with exterior near-infrared PTT and this significantly enhanced its therapeutic effectiveness in cancer treatment. This combination treatment showed a synergistic effect when compared to PTT or PDT alone, resulting in increased therapeutic effectiveness without significant adverse effects on normal organs. Overall, HMME-HA-CNTs were shown to be capable of performing both photodynamic and photothermal treatment simultaneously in future tumor therapy.

## 4. Toxicological Aspect of Carbon Nanotubes

The safety of using CNTs as anti-tumor medication carriers within clinical trials has been a source of concern in recent years [109]. Studies showed that CNTs’ toxic effects are mainly due to their structural resemblance to asbestos fibers [110,111]. Biological persistence, inflammatory response, and malignant mesothelioma are all commonly reported toxic effects of CNTs [112]. The toxicity of MWCNTs was demonstrated by Ursini et al., with results showing that OH- and COOH- functionalized MWCNTs exert toxicity to both A549 and BEAS-2B cancer cells via various mechanisms. In contrast, a number of studies have also shown no toxic effects or significant damage to normal cells [113]. Surface modification, concentration, and aggregation are the main factors which govern the toxicity of CNTs [112]; this is discussed below.

### 4.1. Surface Modification

CNTs have modified surfaces to increase their dispersion, excretion, and biocompatibility [114,115]. CNTs containing anti-tumor medicines have poor water solubility, yet this can be improved by surface modification. Proteins and surfactants added to the CNT surface have been proven to not only increase cancer-cell targeting but also minimize toxicity and boost therapeutic effects [116].

### 4.2. Aggregation

Due to van der Waals forces in a solution, nanoparticles with a small size and large specific surface area have a strong propensity for aggregation [117]. In vivo studies showed that SWCNT toxicity is caused mainly by aggregates rather than individual molecules [118,119]. Reports showed that CNTs that become highly aggregated can become bulky and strong, causing more damage to cells [120].

### 4.3. Shape

Various shapes of CNTs can create different toxic effects. Reports suggest that the inhibition of phagocytosis by SWCNTs can occur due to this, exerting a higher toxic effect in comparison to MWCNTs at a similar dose. In order to investigate thickness-dependent toxicity, Fenoglio et al. demonstrated that thin MWCNTs exert higher toxic effects in comparison to thick MWCNTs [121]. In another study, four different shapes of CNTs were injected into mice, and their mesothelioma effects were investigated. Results showed that, with an increase in curvature, the degree of mesothelioma decreased, i.e., straight-needle shaped CNTs exhibit a higher toxicity and carcinogenicity [122]. In another study, Sakamoto et al. also compared the effect of seven different sizes and shapes of MWCNTs, and the results showed that needle-shaped MWCNTs had a 100% toxic effect in comparison to tangled MWCNTs [123]. Thus, the overall result showed that different shapes display different toxicity levels in cells.

### 4.4. Concentration

CNTs can be translocated to different body organs via blood circulation, where they exert their toxic effect. The concentration of aggregated CNTs is directly proportional to the toxicity level [124]. In this regard, Bottini et al. investigated the effects of 40 μg/mL and 400 μg/mL CNTs on T lymphocytes at different time periods. Results showed that the CNTs did not exert any toxic effect on T cells at a lower concentration of 40 μg/mL, which proves that the toxicity level is directly proportional to the treatment dose. Furthermore, Fanizza et al. demonstrated the concentration-dependent (10, 40 and 100 µg/mL) effects of MWCNTs on BEAS-2B cells [125]. Results showed that after 2 h of exposure, several effects, such as changes in the structure of cellular microvilli, mild herpes development, and reductions in microvilli, were observed. Comet assay results showed that after 4 h of exposure to 40 and 100 µg/mL MWCNTs, significant DNA damage was observed.

### 4.5. Size

Different sizes of CNTs exhibit different toxicity levels to cells. Small CNTs exhibit a high surface area and an enhanced ability to cross the cellular membrane [126]. This can damage the proteins and cellular components and cause dysfunction and death to macrophages [127]. Sohaebuddin et al. demonstrated the effect of MWCNTs with a diameter of less than 8 nm on 3T3 cellular morphology [128]. Exposure of MWCNTs with smaller diameters leads to instability in the lysosomal membrane and induces component release, whereas CNTs with a large dimeter inflict less damage to lysosomes. Furthermore, Martinez et al. investigated the effect of different sized MWCNTs on zebrafish models [129]. Results showed that small MWCNTs exert immunotoxic and neurotoxic effects on larvae, whereas long MWCNTs inflict cardiotoxicity, developmental malformations, and immunotoxicity. Thus, the results clearly indicate that different sized CNTs exert different toxicity effects. Similarly, long SWCNTs also exerted higher toxicity effects than short SWCNTs, further indicating size-dependent toxic effects.

## 5. Advanced Tools to Predict Carbon Nanotube Performance and Applications

Several researchers have utilized a variety of advanced tools to predict the performance and applications of CNTs. Recent research [130] has analyzed the physical properties of CNTs using machine learning techniques. In this context, it is essential to consider the number of parameters, the amount of experimental data, and the algorithms used to model the CNT’s uncontrolled physical properties. Support vector machines, random forests, decision trees, k-nearest neighbors, and artificial neural networks play a crucial role in the analysis of these nanostructures. Using this method, we can also evaluate the electrical, thermal, mechanical, and electronic properties of CNTs. In addition to machine learning techniques, the results of molecular dynamics and density functional theory are required to analyze the electronic, thermal, and electrical properties of CNTs. Machine learning also helps to explain the thermionic and vibrational properties of CNTs by correlating the number of iterations and the detection of defects in carbon nanotubes. CNTs with these types of thermionic and vibrational properties are quite useful for the development of nanosensors. The machine learning and simulation model approach also reduces the cost and time required to analyze the properties of nanomaterials through experimentation. In this way, artificial intelligence and machine learning approaches for analyzing the various properties of nanomaterials are innovative and supplant conventional approaches [131].

## 6. Conclusions

In this review article, we evaluated the recent research and techniques regarding the therapeutic and diagnostic uses of CNTs in various cancer forms. Recent and diverse advancements in CNT-based methods, like the co-administration of CNTs and drugs, irradiation therapy, combined drug treatment, DNA delivery vectors for gene therapy, siRNA-targeted delivery, and CNT-based array biosensor designs that utilize specific antibodies, have been well utilized for cancer therapy. Studying the advantages and disadvantages of using CNTs has led to the development of novel CNT-based drug delivery systems and generally provides better knowledge for a wide range of applications. In order to decrease the toxic effects of CNTs, multi-dimensional investigations need to focus on enhanced biocompatibility, decreased toxicity, and multimodal functionality. Studies into cellular response and cell signaling pathways are also required to establish the biological application of CNTs. However, at all stages of diagnosis and clinical treatment, pharmacogenetic effects should be considered due to population variability and drug sensitivity.

## Figures and Tables

**Figure 1 nanomaterials-12-02283-f001:**
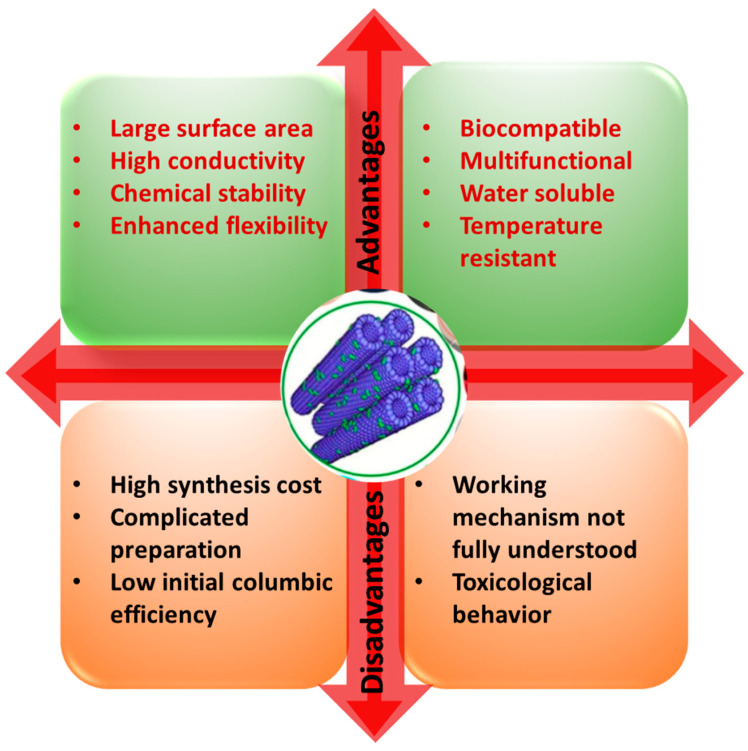
Schematic showing the advantages and disadvantages of carbon nanotubes.

**Figure 2 nanomaterials-12-02283-f002:**
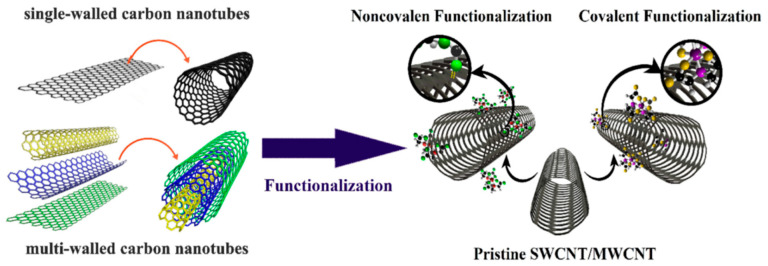
Schematic of the functionalization of SWCNTs and MWCNTs through covalent or noncovalent binding. Reprinted with permission from Life Sciences, Copyright 2020, Elsevier [1].

**Figure 3 nanomaterials-12-02283-f003:**
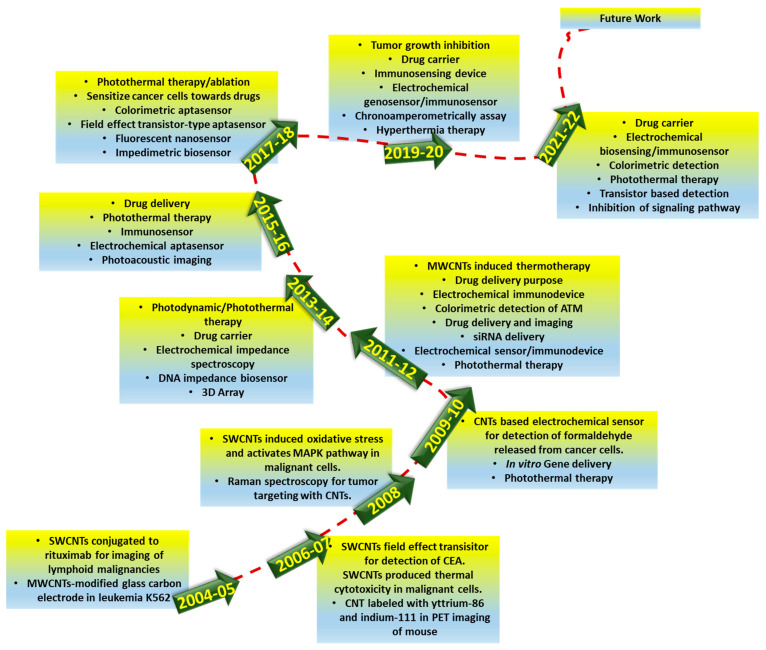
A roadmap for the evolution in the use of carbon nanotubes in cancer targeting and diagnosis.

**Figure 4 nanomaterials-12-02283-f004:**
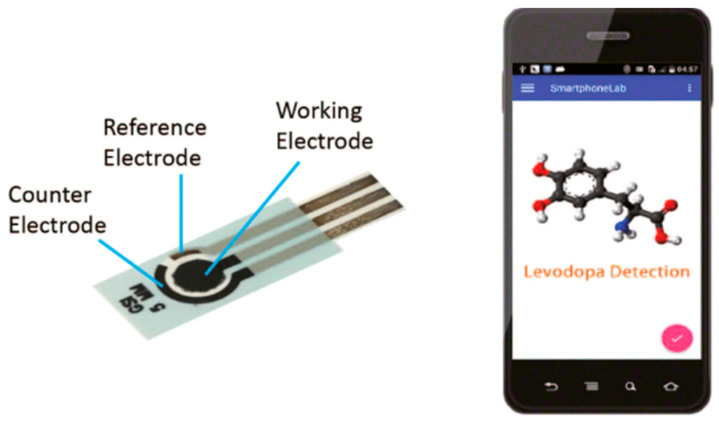
The schematic diagram of the smartphone-based differential pulse amperometry system. Reprinted with permission from Biosensors and Bioelectronics, Copyright 2019, Elsevier [27].

**Figure 5 nanomaterials-12-02283-f005:**
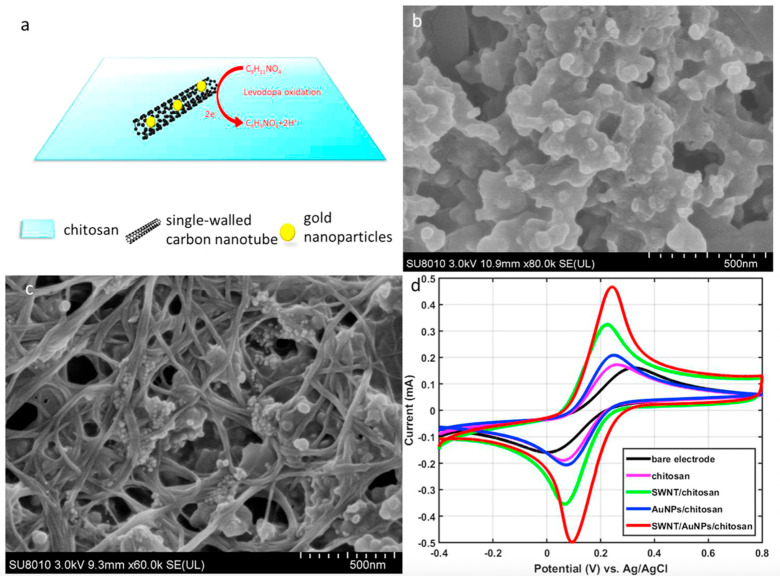
(**a**) Oxidation of levodopa on the surface of the modified working electrode. (**b**) The scanning electron microscope image of the bare screen-printed electrode. (**c**) The scanning electron microscope image of the gold nanoparticle/single-wall carbon nanotube/chitosan-film on the screen-printed electrode. (**d**) Cyclic voltammetry of the redox couple at different electrodes. Reprinted with permission from Biosensors and Bioelectronics, Copyright 2019, Elsevier [27].

**Figure 6 nanomaterials-12-02283-f006:**
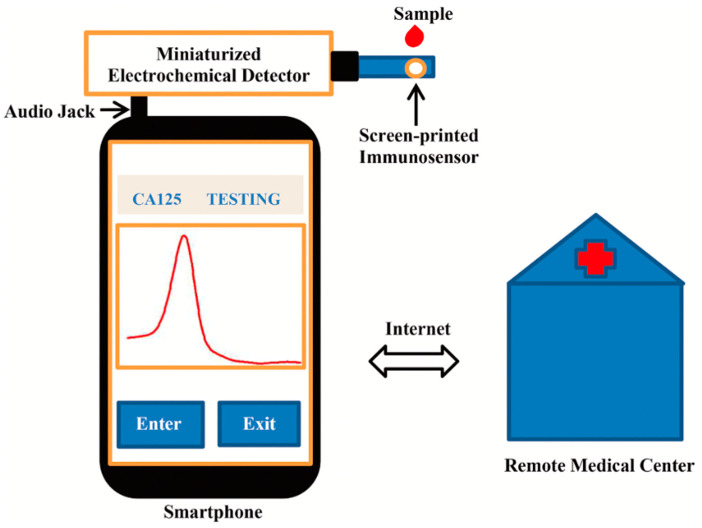
Schematic of the multi-walled carbon nanotube-based electrochemical sensor. Reprinted with permission from Michrochemical Journal, Copyright 2022, Elsevier [28].

**Figure 7 nanomaterials-12-02283-f007:**
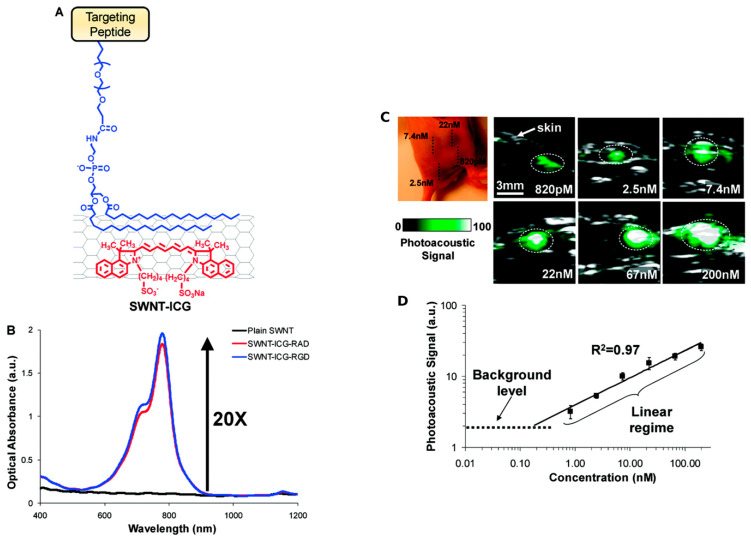
(**A**) Illustration of double CNT functionalization. (**B**) Optical spectra of plain SWNT. (**C**) Photoacoustic (PA) detection of SWNT-ICG in living mice at different concentrations. (**D**) Correlation between the functionalized CNT concentration and the corresponding PA signal. Reprinted with permission from Nano Letters, Copyright 2010, American Chemical Society [43].

**Figure 8 nanomaterials-12-02283-f008:**
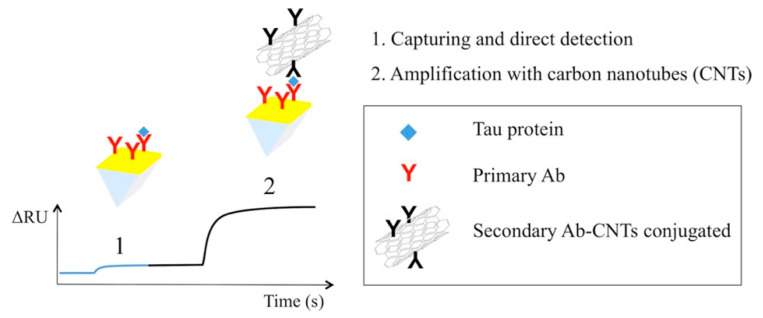
Effect of CNTs during plasmonic biosensing. Reprinted with permission from Biosensors and Bioelectronics, Copyright 2017, Elsevier [52].

**Figure 9 nanomaterials-12-02283-f009:**
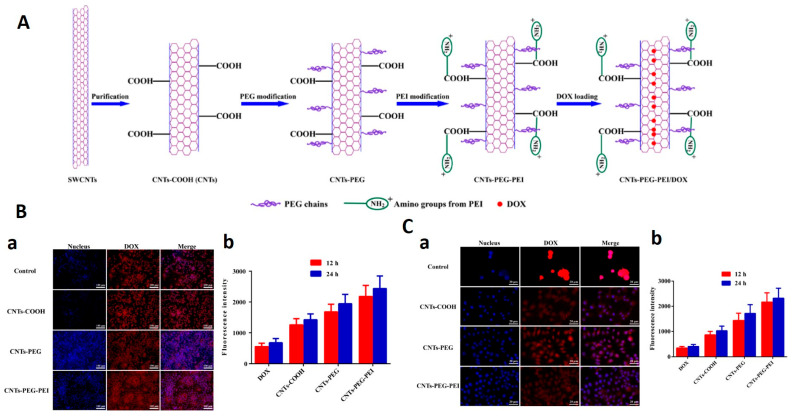
(**A**) Schematic representation of CNT-PEG-PEI nanocarriers and the drug-loading process. (**B**) (**a**) Fluorescence images of MCF-7 cells after treatment with free DOX (control) and different DOX-loaded nanocarrier formulations for 12 h. (**b**) Fluorescence intensity of free DOX and different DOX-loaded nanocarrier formulations (*n* = 3). Student *t*-test was used for statistical analysis. (**C**) (**a**) Confocal microscopy images of MCF-7 cells after treatment with free DOX (control) and different DOX-loaded nanocarrier formulations for 12 h. (**b**) Quantitative fluorescence intensity of free DOX and different DOX-loaded nanocarrier formulations (*n* = 3). Student *t*-test was used for statistical analysis [75].

**Figure 10 nanomaterials-12-02283-f010:**
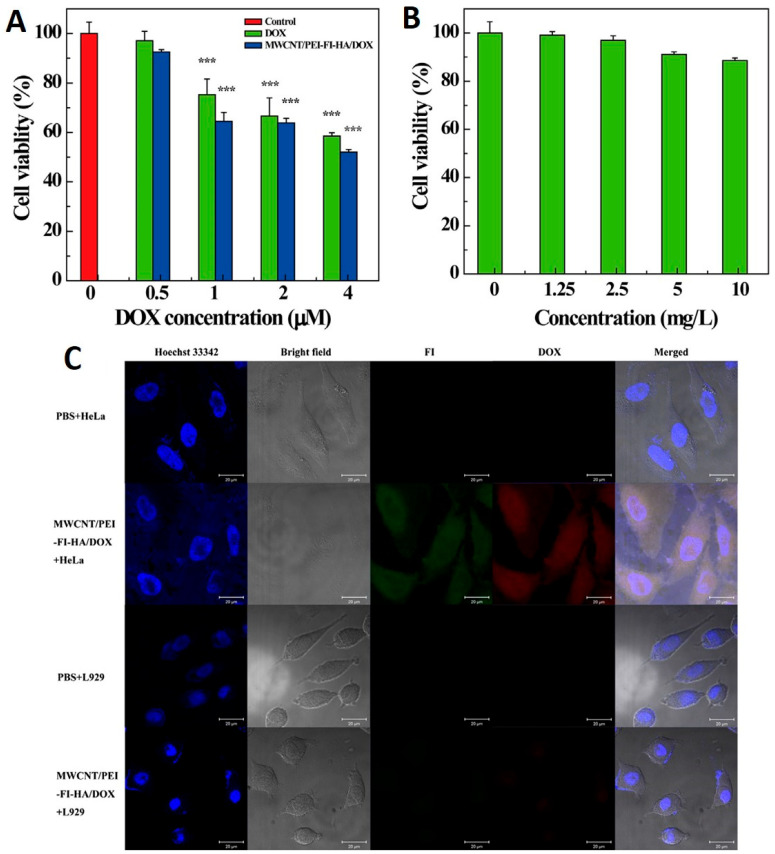
(**A**) MTT viability assay of HeLa cells treated with free DOX and MWCNT/PEI–FI–HA/DOX complexes at the DOX concentrations of 0–4 μM for 24 h, and (**B**) DOX-free MWCNT/PEI–FI–HA at corresponding DOX concentrations of the complexes between 1.25 and 10 mg/L. (**C**) Confocal microscopic images of HeLa and L929 cells treated with MWCNT/PEI–FI–HA/DOX complexes ([DOX] = 2 μM) for 2 h. HeLa and L929 cells treated with PBS were used as controls. The scale bar in each panel represents 20 μm. Reprinted with permission from Carbohydrate Research, Copyright 2015, Elsevier [78]. *** represents the significant level *p* < 0.001.

**Figure 11 nanomaterials-12-02283-f011:**
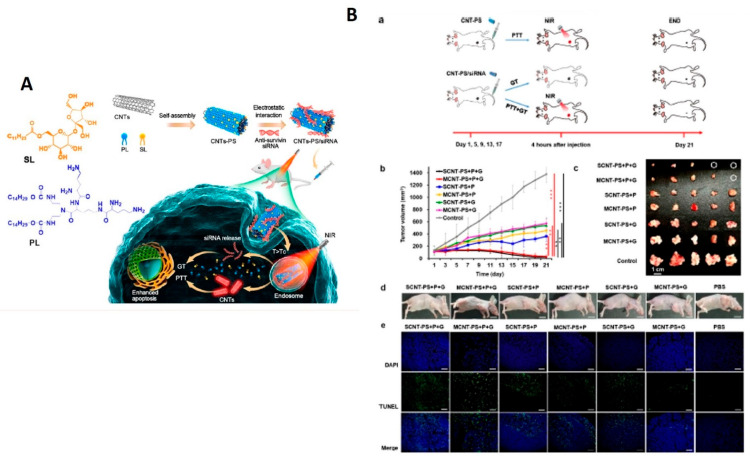
(**A**) Schematic diagram of the temperature-sensitive CNT-PS/siRNA nanoparticle for the synergistic PTT and GT of cancer cells. (**B**) In vivo anti-tumor study. (**a**) Treatment procedure. (**b**) Growth curves of tumors over the course of 21 days after various treatments (*n* = 5 for each group). Each group was treated with the lipid-coated CNT–siRNA complexes ([CNT] = 3 mg/kg, [siRNA] = 1.5 mg/kg) by intravenous injection every 4 days. For groups with PTT treatment, tumor sites were irradiated with an 808 nm NIR laser at 1 W/cm^2^ for 5 min at 4 h after injection. After irradiation, the tumor tissue was found to be 42–45 °C. Mice with PBS injection were used as the control group without any further treatment (mean ± SD, *n* = 5). ** *p* < 0.01; *** *p* < 0.001. (**c**) Representative images of harvested tumors after 21 days of treatment. (**d**) Representative images of mice after the different treatments. One mouse was randomly selected from each group. Scale bar: 1 cm. (**e**) In vivo apoptosis of tumor cells was evaluated by the TUNEL assay (nuclei are stained using DAPI, apoptotic cells are green). Scale bar: 100 μm. Reprinted with permission from ACS Nano, Copyright 2021, American Chemical Society [87].

**Figure 12 nanomaterials-12-02283-f012:**
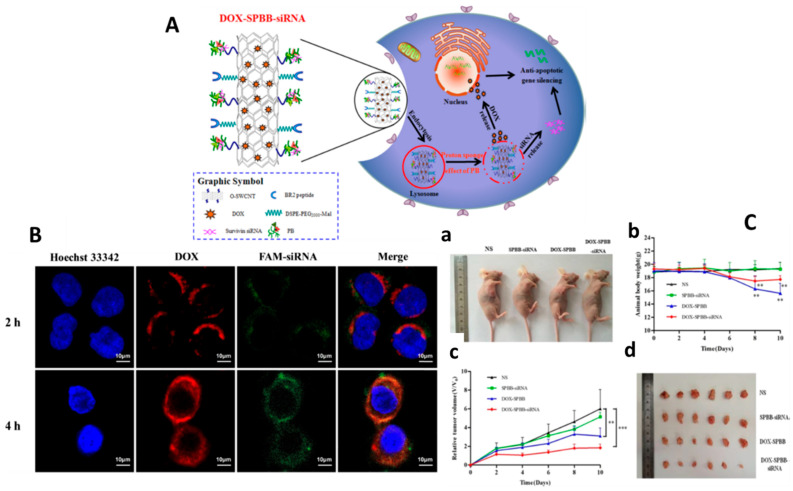
(**A**) Schematic diagram of DOX–SPBB–siRNA nanocarriers in A549 lung cancer cells. (**B**) CLSM images of A549 cells treated with DOX–SPBB–siRNA for 2 and 4 h at given concentrations of DOX (2 μg/mL) and FAM–siRNA (80 nM). Each column from left to right: nuclei stained with Hoechst 33342 (blue); DOX fluorescence in cells (red); FAM signal in cells (green); DOX and FAM-siRNA merged with nucleus. (**C**) In vivo antitumor ability of SPBB loaded with DTX and/or siRNA in A549 tumor-bearing nude mice. (**a**) Appearance of tumor growth. (**b**) Changes in body weight. (**c**) Changes in relative tumor volumes. (**d**) Tumor tissues after being treated for 10 days (*n* = 6, mean ± SD). Reprinted with permission from Applied Materials, Copyright 2019, American Chemical Society [93]. ** *p* < 0.01; *** *p* < 0.001.

**Figure 13 nanomaterials-12-02283-f013:**
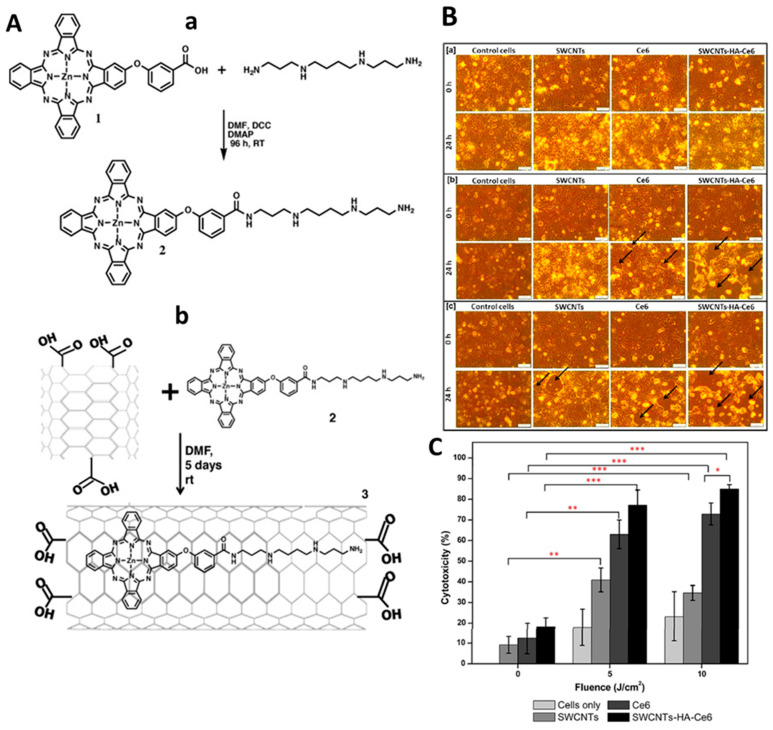
(**A**) Synthetic routes for (**a**) ZnMCPPc-spermine-SWCNT (**2**) and (**b**) ZnMCPPc-spermine-SWCNT (**3**). Reprinted with permission from Synthetic Metals, Copyright 2015, Elsevier [106]. (**B**) Microscopic images of untreated and treated cells. (**a**) Group 1 cells (untreated) at 0 h and 24 h, (**b**) Group 2 cells (5 J/cm^2^) at 0 h and 24 h (**c**) Group 3 (10 J/cm^2^) cells at 0 h and 24 h. Scale bar represents 100 μm. Black arrow indicates the cellular death. (**C**) The cytotoxicity effects of SWCNTs, Ce6, SWCNT-HA-Ce6 on Caco-2 cells determined by LDH assay. Significance is shown as * *p* < 0.05; ** *p* < 0.01; *** *p* 0.001 [108].

**Table 1 nanomaterials-12-02283-t001:** Carbon nanotube-based cancer detection techniques.

Nanocarrier	Cell line/Biomarkers	Linear Range	LoD	Techniques Used	Ref.
SWCNTs	PSA	n.r. ^a^	250 pg/mL	Electrochemical	[53]
SWCNTs	PSA	0.4–40 pg/mL	4 pg/mL	Immunosensing	[54]
MWCNTs	AFP	0.02–2.0 ng/mL	8.0 pg/mL	Immunosensing	[55]
MWCNTs	CEA	0.5–15.0 and 15.0–200 ng/mL	0.01 ng/mL	Immunosensing	[56]
MWCNTs	AFP	0.1–15.0 and 15.0–200.0 ng/mL	0.08 ng/mL	Immunosensing	[57]
MWCNTs	CA 19-9	12.5–270.0 U/mL	8.3 U/mL	Immunosensing	[58]
MWCNTs	hCG	Up to 600 mIU/mL	14.6 mIU/mL	Electrochemical	[59]
MWCNTs	hCG	0.8–500 mIU/mL	0.3 mIU/mL	Electrochemical	[60]
CNTs	PSA	1–100 ng/mL	1.0 ng/mL	Electrochemical	[61]
MWCNTs	CA 125	1.0–30 and 30–150 U/mL	0.36 U/mL	Electrochemical	[62]
CNTs	AFP	1–55 ng/mL	0.6 ng/mL	Immunosensing	[63]
MWCNTs	CA19-9	0–1000 U/mL	n.r. ^a^	Electrochemical	[64]
CNTs	GP73	0–80 ng/mL	58.1 pg/mL	Immunosensing	[65]
CNTs	AFP	0–64 ng/mL	47.1 pg/mL	Immunosensing	[65]
CNTs	AKT2 gene	1 pM–1 μM	2 fM	Electrochemical	[66]
CNTs	CA 125	0.001–0.1 ng/mL and 0.1–30 ng/mL	0.5 pg/mL	Electrochemical	[67]
CNTs	Cyfra 21-1	0.1–10,000 ng/mL	0.5 ng/mL	Fluorescence	[68]
CNTs	HepG2	10–10^5^ cells/mL	5 cells/mL	Electrochemical	[69]

^a^ not reported.

## Data Availability

Not applicable.

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
