# Peer review of "Cancer Targeting and Diagnosis: Recent Trends with Carbon Nanotubes"

_nanomaterials, 2022, doi:10.3390/nano12132283_

Round 1

Reviewer 1 Report

In this paper entitled “Cancer Targeting and Diagnosis: Recent trends with Carbon Nanotubes”, the authors proposed to present the recent progress made in the diagnosis and therapy of cancer utilizing CNTs as mediators of these applications. The review discusses three important points of CNTs utilization: (1) application of CNTs in cancer diagnosis, (2) application of CNTs in cancer therapy, and (3) some toxicological aspects of CNTs. The focus of the paper was on the presentation in detail of advancements in cancer diagnosis and therapy applications based on CNTs covering many strategies for detection (electrochemical, colorimetric, plasmonic, and immunosensing) and therapy (photothermal, photodynamic, drug targeting, gene therapy, and immunotherapy) of several types of cancer. Figures, tables, and images are a strength of this manuscript.

The review is well written and well documented. The 111 cited references are relevant for the review topic, half of them being published within the last 5 years. The paper fit the scope of the journal and it’s suitable for publication in Nanomaterials.

Author Response

Response to Reviewer’s Comments 

Reviewer #1

Comments and Suggestions for Authors: In this paper entitled “Cancer Targeting and Diagnosis: Recent trends with Carbon Nanotubes”, the authors proposed to present the recent progress made in the diagnosis and therapy of cancer utilizing CNTs as mediators of these applications. The review discusses three important points of CNTs utilization: (1) application of CNTs in cancer diagnosis, (2) application of CNTs in cancer therapy, and (3) some toxicological aspects of CNTs. The focus of the paper was on the presentation in detail of advancements in cancer diagnosis and therapy applications based on CNTs covering many strategies for detection (electrochemical, colorimetric, plasmonic, and immunosensing) and therapy (photothermal, photodynamic, drug targeting, gene therapy, and immunotherapy) of several types of cancer. Figures, tables, and images are a strength of this manuscript.

            The review is well written and well documented. The 111 cited references are relevant for the review topic, half of them being published within the last 5 years. The paper fit the scope of the journal and it’s suitable for publication in Nanomaterials.

Response: We are grateful for the positive words from a reputable reviewer. Your feedback and recommendations have helped us make significant improvements to the manuscript.

We thank the worthy reviewer for carefully reviewing the manuscript and providing highly constructive comments for overall improvement of the manuscript.

Reviewer 2 Report

The manuscript ID nanomaterials-1774707 mainly presents a review related to particular studies using advantages of carbon nanotubes in cancer targeting and diagnosis applications. Please see below a list of points to the authors:

1. The authors should clearly describe within the presentation of the report what this review adds beyond previous reports in the topic. You can see for instance: https://doi.org/10.3390/pharmaceutics14040781

2. A roadmap about the evolution in the use of carbon nanotubes for cancer targeting and diagnosis would be welcome.

3. A graphical scheme describing the advantages and disadvantages of carbon nanotubes for cancer targeting and diagnosis should be useful.

4. One major criticism is the absence of a clear discussion involving different results related to SWCNTs and MWCNTs for cancer targeting and diagnosis.

5. The authors are invited to comment about advanced tools to predict carbon nanotube performance and applications. You can see  for instance: https://doi.org/10.1016/j.commatsci.2021.110939

6. It is suggested to split the collective citations in the writing in order to clearly justify the importance of each selected citation to be part of the topic presented.

7. The use of abbreviations must be consistent in the text once they are defined. You can see for instance differences in SW-CNTs and SWCNTs.

8. Only 4 references correspond to 2022, if possible, please update the list of references.

9. The scale bar of figure 8b is missing.

10. A proofreading is mandatory.

Author Response

Manuscript ID: nanomaterials-1774707

Title: Cancer Targeting and Diagnosis: Recent trends with Carbon Nanotubes 

Reviewer #2

Comments and Suggestions for Authors: The manuscript ID nanomaterials-1774707 mainly presents a review related to particular studies using advantages of carbon nanotubes in cancer targeting and diagnosis applications. Please see below a list of points to the authors:

Response: We thank the reviewer for this comment. We have tried our best to address the reviewer’s concerns.

Comment 1:   The authors should clearly describe within the presentation of the report what this review adds beyond previous reports in the topic. You can see for instance: https://doi.org/10.3390/pharmaceutics14040781.

Response: We are thankful to the reviewer for suggestion. We have added the statement showing advantage of our review paper in introduction section.

To date several outstanding articles are published which reported about the involvement of CNTs in cancer therapy. However, no study provides the mechanism based study of CNTs in diagnosis and therapy of cancer. Thus, the present review article provides the deep insight explaining the classified application of CNTs according to their mode of sensing and therapy in cancer cells. Moreover, we have also discussed the toxicological aspects of CNTs regarding their application in biomedical purpose.

Comment 2: A roadmap about the evolution in the use of carbon nanotubes for cancer targeting and diagnosis would be welcome.

Response: We are thankful to the observation. We have added the figure (Figure 3) showing evolution of carbon nanotubes in the introduction section of revised manuscript.

Figure 3: A roadmap about the evolution in the use of carbon nanotubes for cancer targeting and diagnosis.

Comment 3: A graphical scheme describing the advantages and disadvantages of carbon nanotubes for cancer targeting and diagnosis should be useful.

Response: We are thankful to the reviewer for suggestion. We have updated in the revised manuscript  as per given suggestion trying to focus the attention on few aspects, those more related to sensors related to cancer targeting and diagnosis and relative discussion and copied here for your reference. 

Graphical scheme showing the advantage and disadvantage of carbon nanotubes has been added in the introduction section of revised manuscript (Figure 1).

Figure 1: Schematic showing the advantages and disadvantages of carbon nanotubes.

Comment 4: One major criticism is the absence of a clear discussion involving different results related to SWCNTs and MWCNTs for cancer targeting and diagnosis.

Response: Thank you for the observation and suggestion to improve the quality and suitability of the manuscript. As suggested by reviewer, we have added the discussion part in cancer diagnosis section of revised manuscript.

Further, Lv et al. showed the immobilization of bimetallic rhodium@palladium core-shell nanodendrites (Rh@Pd ND) over sulfate functionalized MWCNTs and used it as electrochemical immunosensor for CEA detection. Rh@Pd ND exhibits multiple catalytic site and enhanced surface area, high solubility, and electrical conductivity. In optimal condition, proposed sensor can detect CEA within the linear range of 25 fg/mL-100 ng/ML. Cyclodextrin modification can also be used to improve the immobilization perormace of CNTs.Additionally, carbohydrate antigen (CA) exhibits strong correlation with cancer as its elevated level increases the risk of tumor progression. CA199 can be detected by electrochemical immunosensor composed of antibody conjugated MWCNTs-Fe3O4 dispersed in chitosan. Detection limit has been calculated to be 0.163 pg/mL with linear detection range of 1.0 pg/mL-100 ng/mL.

In another study, Ding et al. demonstrated the antibody functionlaized vertically aligned carbon nanotubes array (VANTA) for detection of ocncoprotein CIP2A, which incuded in various cancers like breast, oral and multiple myloma cancers. VANTA coating has attracted much attention due to unique properties like electrical conductivity, light absorption and chemical inertness. Developed sensor showed the detection limit of 0.24 pg/mL with linear range of 1-100 pg/mL in saliva. Sensor exhibits the higher sensitivity in copariosn to CIP2A enzyme linked immunosorbent assay. Thus, developed senosr paved the way foe rapid and early screening in detection of oral cancer. In another study, Soares et al. also developed the MWCNTs based immunosensor for detection of pancreatic biomarker CA19-9. It also consist of on nanostructured mats of electrospun nanofibers of polyamide 6 and poly(allylamine hydrochloride). Result showed the detection limit of 1.84 U/mL. High sensitivity of sensor may be attributed to irreversible adsorpton among antigen and antibody. Which further confirmed by polarization modulated infrared reflection absoprtion spectroscopy. Proposed sensor has also been tested in real sample of patients blood serum with distinct comcentration of CA19-9. Result showed the accurate detetion with interference from analytes present in biological fluids. Thus, it can be treated as powerful, effective, simple and accurate technique for detection of pancreatic cancer in early stage.

Reports showed that coating of gold over SWCNTs enhances its inherent PA signal, so some researches have developed the “golden nanotubes” (GNTs). 

MWCNTs and SWCNTs with strong NIR absorbance acts as photothermal agent. This storng NIR absorbancy makes nanotubes an excellent contrast medium of PA imaging. Various reports showed the in vitro and in vivo application of SWCNTs in PA imaging. In comparison to control, SWCNTs can offer more than 2 fold and 6 fold signal amplification in thermoacoustic tomography and photoacostic tomogrpahy, respectively. SWCNTs provides the highest contrast signal in comparison to other carbon nanomaterials, graphite, and fullerenes, which makes them an ideal contrast medium candidate for PA imaging.

Thus, it can be conccluded that CNTs exhibiting high NIR absorbance proves to be an excellent contrast agent for PA imaging. Additionally, CNTs combined with different absorptive nanomaterials shows enhanced or mutiplexed PA imaging. Thus, various CNTs based PA imaging probes depends on SWCNTs and MWCNTs for highly enhanced imaging technique.

Fluorescence imaging plays crucial role in medical diagnosis, however low penetration depth limits their further wide application. In order to overcome this problem, researchers have developed advanced fluorescence probes which can be excited at wavelength near to biological transparent NIR window.

Generally, synthezsized SWCNTs did not exhibit fluorescence activity excited under specific wavelength and thus produce dark field images. With the internalizatin of polarization purified SWCNTs, most nanotubes shows significant decrease.

Fluoresence imaging enables the complete tumor removal in depth at a microscopic level.  Ceppi et al. demonstrated reflectance/fluorescence imaging system in ovarian cancer mouse model to both quantify the tumor as well as evaluate the post-operative survival guided by fluorescence image surgeey. In this study, contrast agent composed of SWCNTs conjugated to M13 bacteriophage carrying peptide specific to SPIRC protein (protein overexpressed extracellularly in ovarian cancer). Developed imaging system can detect second near-infrared window fluorescence (1000-1700 nm) and helps in intraoperative tumor debulking by displaying real time video. Author found increased survival of animal tretaed with fluoresence image-guided surgical resection in compariosn to standard surgery. In another study, Lee et al. developed the platelet-derived growth factor (PDGF) aptamer conjugated SWCNTs based NIR optical sensor. Result showed the change in NIR fluorescence of SWCNTs due to the conformational change in aptamer and it can reversibly regulate refolded aptamer-functionalized SWCNTs NIR fluorescence. In another study, Zhang et al. demonstrated the nanocomposite consisting MWCNTs, magnetofluorescent carbon quantum dots for dual modal imaging of cancer cells in mice. 

Raman spectroscopy has been widely used in biomedical diagnostic purpose, as it provides key information regarding chemical composition of cells and tissues. SWNTs exhibits different Raman peaks including tangential G band (~1580 cm-1) and radical breathing mode (RBM 100-300 cm-1). Distinguishment of SWCNTs Raman peak from autofluorescence background can be easily done due to narrow and sharp peak. SWCNTs Raman excitation and scattering photons can reach to NIR region for in vivo imaging.   

Comment 5: The authors are invited to comment about advanced tools to predict carbon nanotube performance and applications. You can see for instance:

https://doi.org/10.1016/j.commatsci.2021.110939

Response: Thank you for the observation and suggestion to improve the quality and suitability of the manuscript. Also, thankful for suggestion of informative recent and relevant article for this comment.

“Several researchers have utilized a variety of advanced tools to predict the performance and applications of carbon nanotubes (CNTs). Recent research [R1] has analyzed the physical properties of CNTs using machine learning techniques. In this context, it is essential to consider the number of parameters, the amount of experimental data, and the algorithms used to model the CNT's uncontrolled physical properties. Support vector machines, random forests, decision trees, K-Nearest Neighbors, and artificial neural networks play a crucial role in the analysis of these nanostructures. Using this method, we can also evaluate the electrical, thermal, mechanical, and electronic properties of CNT. In addition to machine learning techniques, the results of molecular dynamics and density functional theory are required to analyze the electronic, thermal, and electrical properties of CNTs. Machine learning also helps to explain the thermionic and vibrational properties of CNTs by correlating the number of iterations and the detection of defects in carbon nanotubes. CNTs with these types of thermionic and vibrational properties are quite useful for the development of nanosensors. The machine learning and simulation model approach also reduces the cost and time required to analyze the properties of nanomaterials through experimentation. In this way, artificial intelligence and machine learning approaches for analyzing the various properties of nanomaterials are innovative and supplant the conventional approaches [R2].”

[R1] L. E. Vivanco-Benavides, C. L. Martínez-Gonza ́leza, C. Mercado-Zún ̃iga, C. Torres-Torres, Machine learning and materials informatics approaches in the analysis of physical properties of carbon nanotubes: A review, Computational Materials Science, 201, 110939, (2022)

[R2] M. Bahiraei, S. Heshmatian, H. Moayedi, Artificial intelligence in the field of nanofluids: A review on applications and potential future directions, Powder Technol. 353 (2019) 276–301, https://doi.org/10.1016/j.powtec.2019.05.034.

Comment 6: It is suggested to split the collective citations in the writing in order to clearly justify the importance of each selected citation to be part of the topic presented.

Response: Thank you for the observation and suggestion to improve the quality and suitability of the manuscript.

Comment 7: The use of abbreviations must be consistent in the text once they are defined. You can see for instance differences in SW-CNTs and SWCNTs.

Response: Thank you for the observation and suggestion to improve the quality and suitability of the manuscript. Now, we kept SWCNTs at all places in the revised manuscript.

Comment 8: Only 4 references correspond to 2022, if possible, please update the list of references.

Response: Thank you for the observation and suggestion to improve the quality and suitability of the manuscript. As suggested by reviewer, we have added the following 2022 references in the revised manuscript.

  1. Zaboli, M.; Raissi, H.; Zaboli, M. Investigation of nanotubes as the smart carriers for targeted delivery of mercaptopurine anticancer drug. Journal of biomolecular structure & dynamics 2022, 40, 4579-4592, doi:10.1080/07391102.2020.1860823.
  2. Sargazi, S.; Er, S.; Mobashar, A.; Gelen, S.S.; Rahdar, A.; Ebrahimi, N.; Hosseinikhah, S.M.; Bilal, M.; Kyzas, G.Z. Aptamer-conjugated carbon-based nanomaterials for cancer and bacteria theranostics: A review. Chemico-Biological Interactions 2022, 361, 109964, doi:https://doi.org/10.1016/j.cbi.2022.109964.
  3. Murugesan, R.; Raman, S. Recent Trends in Carbon Nanotubes Based Prostate Cancer Therapy: A Biomedical Hybrid for Diagnosis and Treatment. Current drug delivery 2022, 19, 229-237, doi:10.2174/1567201818666210224101456.
  4. Bura, C.; Mocan, T.; Grapa, C.; Mocan, L. Carbon Nanotubes-Based Assays for Cancer Detection and Screening. Pharmaceutics 2022, 14, doi:10.3390/pharmaceutics14040781.
  5. Fan, Y.; Shi, S.; Ma, J.; Guo, Y. Smartphone-based electrochemical system with multi-walled carbon nanotubes/thionine/gold nanoparticles modified screen-printed immunosensor for cancer antigen 125 detection. Microchemical Journal 2022, 174, 107044, doi:https://doi.org/10.1016/j.microc.2021.107044.
  6. Ren, Q.; Zhang, Y.; Ma, S.; Wang, X.; Chang, K.-C.; Zhang, Y.; Yin, F.; Li, Z.; Zhang, M. Low-temperature supercritical activation enables high-performance detection of cell-free DNA by all-carbon-nanotube transistor. Carbon 2022, 196, 120-127, doi:https://doi.org/10.1016/j.carbon.2022.04.068.
  7. González-Domínguez, J.M.; Grasa, L.; Frontiñán-Rubio, J.; Abás, E.; Domínguez-Alfaro, A.; Mesonero, J.E.; Criado, A.; Ansón-Casaos, A. Intrinsic and selective activity of functionalized carbon nanotube/nanocellulose platforms against colon cancer cells. Colloids and Surfaces B: Biointerfaces 2022, 212, 112363, doi:https://doi.org/10.1016/j.colsurfb.2022.112363.
  8. Shahsavar, K.; Alaei, A.; Hosseini, M. Chapter 9 - Colorimetric technique-based biosensors for early detection of cancer. In Biosensor Based Advanced Cancer Diagnostics, Khan, R., Parihar, A., Sanghi, S.K., Eds.; Academic Press: 2022; pp. 153-163.
  9. He, Y.; Hu, C.; Li, Z.; Wu, C.; Zeng, Y.; Peng, C. Multifunctional carbon nanomaterials for diagnostic applications in infectious diseases and tumors. Materials today. Bio 2022, 14, 100231, doi:10.1016/j.mtbio.2022.100231.
  10. Risser, F.; Urosev, I.; López-Morales, J.; Sun, Y.; Nash, M.A. Engineered Molecular Therapeutics Targeting Fibrin and the Coagulation System: a Biophysical Perspective. Biophysical Reviews 2022, 14, 427-461, doi:10.1007/s12551-022-00950-w.
  11. Vivanco-Benavides, L.E.; Martínez-González, C.L.; Mercado-Zúñiga, C.; Torres-Torres, C. Machine learning and materials informatics approaches in the analysis of physical properties of carbon nanotubes: A review. Computational Materials Science 2022, 201, 110939, doi:https://doi.org/10.1016/j.commatsci.2021.110939.

Comment 9: The scale bar of figure 8b is missing.

Response: Thank you for the observation and suggestion to improve the quality and suitability of the manuscript. In the revised manuscript, we have replaced the Fig. 8 with new figure as reference regrading the same was not recent. Due to addition of 2 new figures, now this is a Fig 10.

Comment 10: A proofreading is mandatory.

Response: Thank you for the observation and suggestion to improve the quality and suitability of the manuscript.

We thank the worthy reviewer for carefully reviewing the manuscript and providing highly constructive comments for overall improvement of the manuscript.

Reviewer 3 Report

It was a review paper about the application of CNT based carriers for cancer diagnosis and therapy. Here are some comments on this study that should be considered before publication:

1-     The sample mentioned in line 102-135 “Parkinson's disease is …” is not related to the cancer. Please remove it. The same for figures 2 and 3.

2-     Please introduce all the abbreviations at the first-time usage.

3-     There are some grammatical mistakes in the text that should be corrected.

4-     “… carbon nanotubes attract researchers today, and their clinical applications are intensively studied.” please add related references for this sentence.

5-     Please refer to the Table 1 in main text.

6-     Please add sample for using CNT in colorimetric sensors.

7-     The sample mentioned in section 2.7 is also not related to the cancer.

8-     “In this context, Gu et 332 al. developed the DOX and hydrazinobenzoic acid (HBA) functionalized SWCNTs complex via hydrazine bond and evaluated their cytotoxic effect on HepG2 cancer cells [55]. Result showed that complex exhibit pH dependent drug release rate with maximum release at lower pH 5.5 (tumor cell pH) in comparison to 7.4 pH.” this is related to the hydrazine bond not CNT!

9-     Please improve the quality of figures 9 and 10.

10-  CNTs containing anti-tumor medicines have poor water solubility, which can 603 also be improved by surface modification.147–150.” Please delete 147–150.

11-  Half of the references are old (for more than 6 years ago). Please replace them with update ones.    

Author Response

Manuscript ID: nanomaterials-1774707

Title: Cancer Targeting and Diagnosis: Recent trends with Carbon Nanotubes

Reviewer #3

Comments and Suggestions for Authors: It was a review paper about the application of CNT based carriers for cancer diagnosis and therapy. Here are some comments on this study that should be considered before publication:

Response: We are grateful for the positive words from a reputable reviewer. Your feedback and recommendations have helped us make significant improvements to the manuscript.

Comment 1: The sample mentioned in line 102-135 “Parkinson's disease is …” is not related to the cancer. Please remove it. The same for figures 2 and 3.

Response: We are thankful to the reviewer for suggestion. We have updated in the revised manuscript as per reviewer’s suggestion. In addition to cancer, CNTs are commonly used to diagnose a number of other diseases. These are the fundamental details of CNTs' diverse applications.

Comment 2:   Please introduce all the abbreviations at the first-time usage.

Response: We are thankful to the reviewer for suggestion. We have updated the abbreviations at the first-time usage in the revised manuscript. 

Comment 3:   There are some grammatical mistakes in the text that should be corrected.

Response: We are thankful to the reviewer for suggestion. The entire manuscript has been thoroughly reviewed and all grammatical errors have been eliminated.

Comment 4: “… carbon nanotubes attract researchers today, and their clinical applications are intensively studied.” please add related references for this sentence.

Response: We are thankful to the observation. As suggested we have added following reference to the senstence in the revised manuscript.

He, Y.; Hu, C.; Li, Z.; Wu, C.; Zeng, Y.; Peng, C. Multifunctional carbon nanomaterials for diagnostic applications in infectious diseases and tumors. Materials today. Bio 2022, 14, 100231, doi:10.1016/j.mtbio.2022.100231.

Comment 5: Please refer to the Table 1 in main text.

Response: We are thankful to the observation

Comment 6: Please add sample for using CNT in colorimetric sensors.

Response: We are thankful to the observation. We have modified it as per reviewer’s suggestions.

He, Y.; Hu, C.; Li, Z.; Wu, C.; Zeng, Y.; Peng, C. Multifunctional carbon nanomaterials for diagnostic applications in infectious diseases and tumors. Materials today. Bio 2022, 14, 100231, doi:10.1016/j.mtbio.2022.100231.

Comment 7: The sample mentioned in section 2.7 is also not related to the cancer.

Response: We are thankful to the observation. This is a cutting-edge method, which will be discussed further to enlighten readers about the work of the future.

Comment 8: “In this context, Gu et 332 al. developed the DOX and hydrazinobenzoic acid (HBA) functionalized SWCNTs complex via hydrazine bond and evaluated their cytotoxic effect on HepG2 cancer cells [55]. Result showed that complex exhibit pH dependent drug release rate with maximum release at lower pH 5.5 (tumor cell pH) in comparison to 7.4 pH.” this is related to the hydrazine bond not CNT!

Response: We are thankful to the meticulous observation. In this, SWCNTs complex has been used via hydrazine bond and evaluated their cytotoxic effect on HepG2 cancer cells.

Comment 9: Please improve the quality of figures 9 and 10.

Response: We are thankful to the observation. We have replaced the figures 9 and 10 with high-resolution figures.

Comment 10: “CNTs containing anti-tumor medicines have poor water solubility, which can 603 also be improved by surface modification.147–150.” Please delete 147–150.

Response: We are thankful to the observation. This is typo error and we have removed it from the revised manuscript.

Comment 11: Half of the references are old (for more than 6 years ago). Please replace them with update ones.

Response: We are thankful to the observation. We have updated the list of references with more recent articles in the revised manuscript.

We thank the worthy reviewer for carefully reviewing the manuscript and providing highly constructive comments for overall improvement of the manuscript.

Round 2

Reviewer 2 Report

In my opinion the reviewed version of this manuscript can be accepted for presentation in present form. I noticed that all the points raised in the review stage have been successfully addressed. Brightly, the authors are presenting an original topic, well discussed, and with a beautiful presentation.

Reviewer 3 Report

Thanks for addressing the comments.